# Identification of *GA2ox* Family Genes and Expression Analysis under Gibberellin Treatment in Pineapple (*Ananas comosus* (L.) Merr.)

**DOI:** 10.3390/plants12142673

**Published:** 2023-07-17

**Authors:** Wenhui Zhu, Jingang Qi, Jingdong Chen, Suzhuo Ma, Kaichuang Liu, Han Su, Mengnan Chai, Youmei Huang, Xinpeng Xi, Zhuangyuan Cao, Yuan Qin, Hanyang Cai

**Affiliations:** 1College of Life Sciences, Fujian Agriculture and Forestry University, Fuzhou 350002, China; 3200130066@fafu.edu.cn (W.Z.); 1200102024@fafu.edu.cn (J.Q.);; 2College of Agriculture, Yangtze University, Jingzhou 434025, China

**Keywords:** pineapple, *GA2ox* family, phylogenetic analysis, synteny analysis, expression profiles

## Abstract

Gibberellin (GAs) plays an important regulatory role in the development and growth of pineapple (*Ananas comosus* (L.) Merr.). Bioinformatics was used to confirm the differential expression of GA2 gibberellin oxidase gene *AcGA2oxs* in the pineapple genome, which laid the foundation for exploring its role in pineapple. In this study, 42 *GA2ox* genes *(AcGA2oxs)* were identified in the pineapple genome, named from *AcGA2ox1* to *AcGA2ox42*, and divided into four groups according to phylogenetic analysis. We also analyzed the gene structure, conserved motifs and chromosome localization of AcGA2oxs. AcGA2oxs within the same group had similar gene structure and motifs composition. Collinear analysis and cis-element analysis provided the basis for understanding the evolution and function of *GA2ox* genes in pineapple. In addition, we selected different tissue parts to analyze the expression profile of *AcGA2oxs*, and the results show that 41 genes were expressed, except for *AcGA2ox18*. *AcGA2ox18* may not be expressed in these sites or may be pseudogenes. qRT-PCR (real-time fluorescence quantitative PCR) was used to detect the relative expression levels of the *GA2ox* gene family under different concentrations of GA3 treatment, and it was found that *AcGA2ox* gene expression was upregulated in different degrees under GA3 treatment. These results provide useful information for further study on the evolution and function of the *GA2ox* family in pineapple.

## 1. Introduction

Gibberellins (GAs) is an important hormone in plant growth and development which plays an important role in regulating plant growth and development, including seed germination, stem elongation, root growth, leaf growth, flowering induction and fruit development [1,2,3,4,5]. In recent years, with the in-depth study of gibberellin synthesis and metabolic pathways, almost all gibberellin-related genes have been cloned in large numbers. Although at least 136 kinds of gibberellins with different structures have been isolated and identified from a variety of organisms [6], only a very small number of gibberellins with biological activity have been found through studies (GA1, GA3, GA4, GA5, GA6 and GA7) [7].

The *GA2ox* genes belongs to the 2OG-Fe(II)oxygenase subfamily, and all members of this subfamily contain a 2OG-Fell_Oxy domain. *GA2ox* are important genes in the gibberellin metabolic pathway, which is mainly involved in GA inactivation metabolism. The inactivation metabolism of GA is mainly 2β-hydroxylation. The most typical deactivation reaction is 2-β-hydroxylation catalyzed by a class of 2ODDs oxidase (GA2oxs). The initially identified GA2ox was based on C19GA [8,9], including the biologically active GA and its direct precursors GA9 and GA20 [10], which belonged to class I or Class II 2ODDs oxidase according to phylogenetic relationships [10]. Later, a novel GA2ox oxidase that accepts only C20GA as a substrate was reported [11]. These enzymes are classified as Class III 2ODDs oxidase and reduce the amount of biologically active gibberellin by consuming the precursors of synthetic bioactive gibberellin, such as GA12 and GA53 [12].

The *GA2ox* gene has been identified and cloned in many plants, such as rice (*Oryza sativa*) [13], *Arabidopsis thaliana* [14], *Phyllostachys heterocycle* and tobacco (*Nicotiana tobacum*) [15,16]. Overexpression of the *GA2ox* gene can degrade bioactive gibberellin in plants and lead to dwarfing phenotype [17]. For example, in wheat, ectopic expression of the *PcGA2ox1* gene in soybean significantly reduces GA content, which led to lower plant height [18]. In rice, overexpression of the *OsGA2ox1* gene resulted in a severe reduction in plant height [19], and overexpression of the *OsGA2ox6* or *OsGA2ox9* gene resulted in a moderate reduction in plant height [20,21].

Pineapple (*Ananas comosus* (L.) Merr.), widely cultivated in tropical and some subtropical regions, is an important tropical fruit in horticultural production in the world [22]. Pineapple has a high economic value, and pineapple cultivation for local agricultural development is of great significance.

In recent years, the application of plant growth regulators to control the growth and development of pineapple has become a necessary means for the intensive cultivation and development of pineapple. Gibberellin (GA), as an important hormone, has the function of promoting growth. The most obvious biological activity of GA is to promote the elongation of plant cells and make plants grow taller. It can break the dormancy of seeds, tubers and roots and promote their germination. It can stimulate fruit growth, improve seed setting rate or form seedless fruit [23]; promote the early flower bud differentiation of some plants that need low temperature to pass the growth stage and partially compensate the role of sunshine, so that some plants can also bolt and flower under short-day conditions; and induce the formation of α-amylase and accelerate the hydrolysis of stored substances in endosperm cells. As previously reported, we found that exogenous gibberellin can cause browning inside pineapple, and GA2ox can play an important role in reducing GA levels and reducing internal browning (IB) [24]. However, the *GA2ox* gene family has not been systematically analyzed in pineapple. Therefore, our analysis focused on the identification of the *AcGA2ox* genes and the characterization of encoding the *AcGA2ox* family.

In this study, we identified 42 *AcGA2ox* genes, divided them into four subgroups, and analyzed their gene and protein structures, protein motifs, chromosome distribution and expression profiles. Our results provide a relatively complete profile of the GA2ox gene family in pineapple. This may facilitate further functional analysis of each member and facilitate the improvement of pineapple varieties through gene transfer technology.

## 2. Results

### 2.1. Identification and Physicochemical Property Analysis of AcGA2ox Gene Family

The protein sequences of nine *Arabidopsis thaliana*
*GA2ox* genes were downloaded from Tair, the nine *AtGA2ox* genes were used as seed sequences and the pineapple whole-genome protein sequence file was used as database, with e value set at 1e-20, and the possible *AcGA2ox* members were identified by Blast. The 2OG-FeII_Oxy domain HMM matrix of GA2ox was downloaded from the Pfam website, and the AcGA2ox protein with this domain was searched in the database by HMMER3.3.2 software. The protein members identified by Blast and HMM were identified, and incomplete domain sequences were removed by CD-Search tool. Finally, 42 *AcGA2ox* members were identified, which were named *AcGA2ox1-42* according to gene ID sequence. Forty-two members were mapped to nineteen chromosomes (Figure 1), among which chromosomes LG3 and LG4 contained the most GA2ox members (five), and three members were found on LG2, LG6, LG9, LG12 and LG25. No AcGA2ox copies were identified on LG14, LG15, LG16, LG19, LG20 and LG24, and most chromosomes contain one or two members (Table 1). The coding length of the AcGA2ox protein is 150–526 amino acids, the predicted molecular weight is 16.87–118.47 kD and the isoelectric point (pI) is 4.93~8.46. The prediction results show that AcGA2ox41 and AcGA2ox7 exhibit hydrophobicity, and their average hydrophilic values (GRAVY) are 0.136 and 0.146, respectively. Other proteins are hydrophilic, with GRAVY ranges from −0.583 to −0.031 (Appendix A).

### 2.2. Construction and Analysis of Phylogenetic Tree of AcGA2ox Members

In order to elucidate the evolutionary relationship between the pineapple GA2ox protein and the model plant *Arabidopsis thaliana*, a phylogenetic tree was constructed by NJ method. According to sequence similarity, 9 AtGA2ox members and 42 AcGA2ox members were grouped into two clades, namely Clade I and Clade II. The Clade II clades can be further classified into three subclasses: Clade II-A, Clade II-B and Clade II-C (Figure 2). Notably, Clade I and Clade II-A did not cluster to the Arabidopsis GA2ox protein, and Clade I accounted for a relatively large proportion in pineapple, containing 25 members. Clade II-A contains eight members. Clade II-B has three members and Clade II-C has six members (Table 2).

According to the phylogenetic tree, among the nine GA2ox members in *Arabidopsis*, *AtGA2ox1* is related to *AtGA2ox2*, *AtGA2ox3*, *AtGA2ox4* and *AtGA2ox6*. *AtGA2ox7* is closely related to *AtGA2ox8*, *AtGA2ox9* and *AtGA2ox10* (Figure 3). Interestingly, 33 members of GA2ox in pineapple have no direct homologous genes in *Arabidopsis*, which may be a new subclass formed in pineapple. These 33 AcGA2ox were clustered into Clade II-A and Clade II-A, suggesting that some Clade II-A may have similar biological functions with Clade II-B and Clade II-C, while Clade I may not. These results suggest that the GA2ox gene in pineapple may have a similar function to that in *Arabidopsis* and may also derive some new biological functions.

### 2.3. Collinearity Analysis of AcGA2ox Genes

The collinearity analysis of *GA2ox* genes among the 3 model plants of pineapple, *Arabidopsis* and rice showed that there were 3 pairs of collinearity genes between pineapple and *Arabidopsis*, and 19 pairs of collinearity genes between pineapple and rice (Figure 3). The intraspecies collinearity analysis showed that 11 pairs of *GA2ox* existed in the pineapple genome (Figure 4). The difference in the number of homologous gene pairs between pineapple and the two model plants may be due to the small *Arabidopsis* genome. There are few collinear gene pairs, which indicates that the retention rate of *GA2ox* gene is not high during the evolution of pineapple species.

### 2.4. Gene Structure and Cis-Acting Element Analysis of Conserved Motif of AcGA2ox Gene Family

The 42 AcGA2ox proteins contained 10 conserved moTIFs (Motif1~10, Appendix A), and Motif1 existed in all Clades. Motifs 2, 3, 5 and 7 were widely present in Clade II, while some members of Clade I were not. Motifs 4 and 6 were widely present in Clade II-B and Clade II-C, while some members of Clade I and Clade II-A were absent. Motif 9 was widely present in Clade II-B but not in Clade I, Clade II-A or Clade II-C. Motifs 8 and 10 were present in some members of Clade I and Clade II-A but not in Clade II-B and Clade II-C (Figure 5A). All 42 AcGA2ox proteins contain the 2OG-FeII_Oxy domain, AcGA2ox19 contains 3 of this domain, AcGA2ox37, AcGA2ox12, AcGA2ox32 and AcGA2ox14 contain 2 of this domain, and the other proteins contain only 1 of this domain (Figure 5B). TBTools software was used to predict the cy-acting elements in 42 *AcGA2ox* gene promoter regions (Figure 5C), and the number of cy-acting elements related to hormones, meristem, stress and circadian rhythm was analyzed to further explore the function of *AcGA2ox*. In this study, our results show that stress-related action elements are prevalent and relatively abundant on the promoter of *AcGA2ox*. Hormone-related action elements such as methyl jasmonate, gibberellin and abscisic acid were compared. Overall, the amount associated with abscisic acid was relatively small, with only the *AcGA2ox10* and *AcGA2ox30* elements associated with wound response, and only the *AcGA2ox28* and *AcGA2ox35* genes were associated with cell cycle response. Nearly half of the genes did not contain gibberellin-responsive elements. Many genes contain abiotic stress elements, such as low temperature, drought and other stress response components. Among the 42 *AcGA2ox* genes, the number of CDS (Coding sequence) ranged from 2 to 11 (Figure 5D). *AcGA2ox1*, *AcGA2ox18*, *AcGA2ox20*, *AcGA2ox29*, *AcGA2ox33*, *AcGA2ox39* and *AcGA2ox40* had only 2 CDS, while AcGA2ox4 had the largest number of CDS, reaching 11. Some closely related members showed consistency in gene structure, but some showed abundant variation, and no significant difference was found in gene structure among different subclasses, which indicated that the selection pressure of the gene structure was relatively relaxed and the variation was abundant.

### 2.5. The AcGA2ox Gene Family Showed a Tissue-Specific Expression Pattern

To further analyze the biological function of AcGA2oxs, we collected biological samples of various tissues of pineapple (ovule, sepal, stamen, petal, root, flower, leaf and fruit). The expression of 42 *AcGA2ox* genes in these tissues was obtained by RNA-Seq technique, and the expression data heat map was constructed (Figure 6). The expression of *AcGA2ox* was found to be low in half of these tissues. However, it is worth noting that *AcGA2ox1*, *AcGA2ox13*, *AcGA2ox2*, *AcGA2ox10*, *AcGA2ox23*, *AcGA2ox22*, *AcGA2ox41* and *AcGA2ox7* had high expression levels in the above tissues. The expression of *AcGA2ox9* was low only in ovule but high in other tissues. The expression of *AcGA2ox39* was lower in petal and fruit but higher in other tissues. The expression of *AcGA2ox35* was low in flower and leaf but high in other tissues. *AcGA2ox4*, *AcGA2ox24*, *AcGA2ox37*, *AcGA2ox20*, *AcGA2ox15*, *AcGA2ox3*, *AcGA2ox6*, *AcGA2ox32*, *AcGA2ox40*, *AcGA2ox25*, *AcGA2ox14*, *AcGA2ox38* and *AcGA2ox17* were highly expressed in some tissues. *AcGA2ox18* was not expressed in any of the tested samples, possibly because it is a pseudogene or has a special spatiotemporal expression pattern not detected in our library. Therefore, it is speculated that *AcGA2ox* may play an important role in various growth and development stages of pineapple through the specific expression of some members. 

The reliability of the transcriptome data was further validated by an RT-PCR assay, which validated 7 representative samples of 10 selected *AcGA2ox* genes (Figure 7). Some genes were highly expressed in the detected tissues, including three genes (*AcGA2ox7*, *AcGA2ox9* and *AcGA2ox10*) in stamens. The expression levels of three genes (*AcGA2ox7*, *AcGA2ox10* and *AcGA2ox13*) in sepals and one gene (*AcGA2ox15*) in petals were higher. Some genes are only expressed in certain locations, for example, *AcGA2ox5* and *AcGA2ox9* are only detected in pistil, stamen and flower.

### 2.6. Analysis of GA Hormone-Induced Expression Pattern of AcGA2ox Gene Family

In order to explore the potential role of *AcGA2ox* in the interaction with hormones, we analyzed the gene expression of some members of the *AcGA2ox* family by q-PCR. According to previous studies, some *GA2ox* genes are regulated by exogenous hormones, such as those of *medicago truncatula* [25] and corn [26]. In order to study the effect of exogenous gibberellin on the expression of the *AcGA2ox* gene, 10 *AcGA2ox* genes were selected from the leaves to study their response to exogenous GA3. Three gradients of gibberellin concentration were used to study the effect of exogenous gibberellin on the expression the of *AcGA2ox* gene. The expression profiles of GA3 at the concentrations of 25 mg/L, 50 mg/L and 100 mg/L were analyzed by RT-PCR (Figure 8). Generally speaking, we found that the expression of these genes was upregulated to different degrees. The expression of *AcGA2ox15*, *AcGA2ox39* and *AcGA2ox41* increased first and then decreased; *AcGA2ox5* decreased first and then increased; and *AcGA2ox38* showed a fluctuation between decline, rise and fall. Interestingly, different concentrations of GA3 also had different effects on the expression of the same gene. For example, *AcGA2ox9*, *AcGA2ox22* and *AcGA2ox10* were upregulated and then downregulated at 25 mg/L, and *AcGA2ox9* returned to its original expression level after 12 h. These genes were upregulated at 100 mg/L, which indicated that some genes may need a longer expression time to eliminate the effect of higher concentrations of GA3. The highest expression level of *AcGA2ox39* was 114.8 times that of the control. The expression of *AcGA2ox5* was the least affected by exogenous GA3.

### 2.7. Subcellular Localization of AcGA2ox Proteins

Several representative proteins (AcGA2ox5, AcGA2ox7, AcGA2ox10, AcGA2ox13, AcGA2ox9, AcGA2ox38 and AcGA2ox22) from each subgroup were selected for subcellular localization studies to explore the expression location of AcGA2ox protein in cells. The Ubi:AcGA2ox:GFP carrier and the control Ubi:GFP were injected into the epidermal cells of Ben’s tobacco leaves (Figure 9). Subcellular localization analysis showed that AcGA2ox5, AcGA2ox7, AcGA2ox10 and AcGA2ox13 were located in the nucleus and cell membrane of tobacco epidermal cells. AcGA2ox9, AcGA2ox38 and AcGA2ox22 were localized only in the cell membrane. This is consistent with previous results [27].

## 3. Discussion

The *GA2ox* family plays an important role in the synthesis and metabolism of gibberellin in plants and belongs to the gibberellin oxidase gene family, which also includes *GA3ox* and *GA20ox* [28]. It has been reported that members of the *GA2ox* family play a crucial role in the growth and development of various plants [29,30,31,32]. Previous studies have identified 12 *GA2ox* family members in maize [26], 11 *GA2ox* family members in grape and 10 GA2ox family members in soybean [33,34]. Eleven and nine members of the GA2ox family have been identified in rice and Arabidopsis, but the pineapple *GA2ox* family has not been identified. Pineapple is a tropical fruit with great economic value and research value, so it is necessary to study the role of the GA2ox family in pineapple.

In this study, bioinformatics was used to search for gene members with similar 2OG-Fell_Oxy domain in the pineapple genome. A total of 42 *AcGA2ox* genes were identified, some of which may be pseudogenes or duplicate events. The degree of genetic differentiation is still much higher than that of maize, grape, soybean, rice and *Arabidopsis*, and the diversity of gene structural patterns means that the process of generating new parallel sequences is more complex. Members of the pineapple GA2 oxidase gene with longer introns appear, such as *AcGA2ox14*, *AcGA2ox19*, *AcGA2ox11*, *AcGA2ox32* and *AcGA2ox37*. Long introns can arise from unequal crossings, while random insertion of a retrotransposon can produce a new gene. In order to further reveal the phylogeny of the pineapple *GA2ox* family, a phylogenetic tree of the pineapple *GA2ox* family was constructed, and 42 members of the pineapple *GA2ox* family were divided into 4 groups (Figure 2). 

Analysis of promoter cis-acting elements showed that *AcGA2ox* contained some cis-acting elements related to hormones, meristem, stress and cell cycle regulation. These genes are also affected by GA regulation. Interestingly, a considerable number of gene promoters do not contain gibberellin-related elements, suggesting that their function may be to regulate changes in endogenous GA content through other factors on the GA signaling pathway, thereby regulating GA levels in plants.

Gene expression patterns can preliminarily predict gene function. Transcriptome results showed that only a small part of the AcGA2ox gene was highly expressed in plants, because GA2ox, as an enzyme for the digestion of gibberellin activity, can only increase the expression level when gibberellin is overpresent in plants. qRT-PCR results show that three genes (*AcGA2ox10*, *AcGA2ox13* and *AcGA2ox15*) showed high expression levels in selected pineapple tissues, suggesting that GA2ox may play a role in regulating gibberellin concentration during the development of pineapple. Some AcGA2ox genes are highly expressed in pistils, stamens and sepals, such as *AcGA2ox7*, *AcGA2ox10*, *AcGA2ox13*, etc., indicating that these genes play an important role in flower formation and the reproductive stage of plants. In addition, some *AcGA2ox* genes are expressed in different tissues or at different stages, indicating that these genes may be more stable than genes expressed only at a certain stage in specific tissues or organs [35]. The tissue expression pattern indicated that AcGA2oxs play essential roles in pineapple reproductive development, as shown in (Figure 10). It also provides an important foundation for understanding the roles of GA2ox genes in regulating the leaf development and root growth in pineapple [8,36].

The relationship between *GA2oxs* and GA in pineapple was studied by qRT-PCR method after different concentrations of gibberellin were treated. The results show that the expression of these genes was upregulated after GA3 treatment for a period of time, because GA2 oxidase is a key enzyme in the degradation process of GA, which can inactivate bioactive GAs, its precursors and other intermediates in plants, thus maintaining the balance between biologically active GAs and intermediates in plants [37,38]. Generally, the response time of pineapple *GA2oxs* to gibberellin at different concentrations was different, and the overall trend was that the expression of GA3 at low concentration reached the peak earlier than that at high concentration and then began to decline, indicating that the deactivation of exogenous GA at higher concentration required more expression of *GA2oxs* in pineapple.

## 4. Materials and Methods

### 4.1. Identification of GA2ox Gene Family Members and Analysis of Physical and Chemical Properties of Pineapple Protein

Using Tair (https://www.arabidopsis.org/ accessed on 22 July 2022) to retrieve the Arabidopsis (*Arabidopsis thaliana*) *GA2ox* genes, its gene ID and protein sequences were reported. The protein sequence, genomic DNA sequence, gene CDS sequence and GFF annotation information of Arabidopsis thaliana and pineapple (*Ananas comosus* (L.) Merr.) were downloaded from the Phytozome database (https://phytozome-next.jgi.doe.gov/ accessed on 22 July 2022). The obtained AtGA2ox protein sequence was used as the seed sequence, and the whole protein sequence of pineapple was used to build a library. BLAST was carried out in the library, the E value was set to 1e-20 and the copies were deleted. From the PFAM database (https://www.ebi.ac.uk/interpro, accessed on 22 July 2022), the matrix of the 2OG-FeII_Oxy domain was downloaded from (http://pfam.xfam.org/, accessed on 22 July 2022), and the *AcGA2ox* gene was retrieved by HMMER 3.3.2 software [39]. Combining the results obtained from BLAST and HMMER, AcGA2ox members were identified. Using the CD-Search tool (https://www.ncbi.nlm.nih.gov/Structure/bwrpsb/bwrpsb.cgi/, accessed on 22 July 2022) to verify the integrity and reliability of the structural domain (E value < 0.01), the incomplete structure domain sequence was removed. Finally, the number of *AcGA2ox* genes in the family was determined. The ExPASy ProtParam tool was used on the server (https://web.expasy.org/protparam/, accessed on 22 July 2022) to predict GA2ox proteins’ isoelectric point (pI), molecular weight (MW) and other physical and chemical properties.

### 4.2. Construction of Phylogenetic Tree of GA2ox Gene Family

Multiple sequence alignment and phylogenetic analysis were used to detect the evolutionary relationship between Arabidopsis and pineapple GA2ox protein. In MEGAX 7.0 software [40], the neighbor join (NJ) method was used to generate the system tree, and multiple sequences were compared with default parameters. The bootstrap number was set to 1000 to estimate the branch length. The tree is visualized using iTOL (https://itol.embl.de/, accessed on 22 July 2022).

### 4.3. Prediction of Gene Structure and Cis-Acting Element of Protein Conserved Domain

The MEME website (http://meme-suite.org/tools/meme, accessed on 22 July 2022) was used to predict the conserved Motif of the AcGA2ox protein, and the number of motifs was set to 10. The gene structure of AcGA2ox was analyzed by TBTools [41]. The upstream 2 kb of the start codon of the *AcGA2ox* gene family was extracted as the promoter sequence, and the composition of its cis-acting elements was analyzed by TBTools to understand its abiotic stress response specificity. The above analysis results were visualized using TBTools.

### 4.4. Chromosome Location and Collinearity Analysis

Using the downloaded GFF annotation information, the *GA2ox* genes present on the definite chromosomes of pineapple were located on their respective chromosomal locations and visualized using the MG2C website [42]. MCScanX was used to analyze the collinearity between pineapple genome and Arabidopsis and rice [43], and TBTools was used for visualization.

### 4.5. Expression Analysis of GA2ox Gene Family in Different Tissues of Pineapple

By studying the transcription levels of various tissues during the development of pineapple, we hope to find the important role of AcGA2ox in plant growth. Transcriptome data came from the previous study of our research group [35,38,44]. Transcripts with consistent sequences were obtained by local BLAST comparison, then corresponding FPKM values were calculated and expression heat maps were drawn by TBTools.

### 4.6. Plant Material and Sample Preparation

The pineapple variety is “*MD2*”, provided by Qin Yuan’s research group (Genomics and Biotechnology Center, Fujian Agriculture and Forestry University, Fuzhou, China). It is grown in a greenhouse at 25 °C, with 16 h light and 8 h dark treatment and relative humidity at 70%, lasting for one month. When the plants reached the flowering stage, different floral organs (i.e., ovule, sepal, stamen, petal, flower and fruit) were collected, including root and leaf. Gibberellin treatment group treated in the following ways: 25 mg/L GA3, 50 mg/L GA3 and 100 mg/L GA3. Pineapple leaf samples were collected 1, 3, 6 and 12 h after treatment. One-month-old pineapple plants without any treatment were used as controls. All samples were harvested, snap-frozen using liquid nitrogen and kept at −80 °C until RNA extraction. Three biological replicates of each sample were collected for analysis.

### 4.7. RNA Isolation and qRT-PCR Analysis

RNA kit (OMEGA, Shanghai, China) extracted total RNA. All samples were reverse-transcribed into cDNA with 1 μg RNA by a reverse transcription reagent (Vazyme HiScript III All-in-One RT SuperMix Perfect for qPCR). For the differences between primer design and respective *AcGA2ox* sequences, all primers were synthesized by Shenggong Biological Company (Fuzhou, China), and the primer sequences are shown in (Appendix A). The qRT-PCR reaction system was designed to be 20 μL, including 10 μL mix (Taq Pro Universal SYBR qPCR Master Mix, Shanghai, China), 6 μL water, 1 μL F-terminal primer, 1 μL R-terminal primer and 2 μL template. The RT-PCR reactions program was completed with the following conditions: 95 °C for 30 s; 40 cycles of 95 °C for 5 s and 60 °C for 34 s; and 95 °C for 15 s [45,46]. The analyses were confirmed in triplicate. The relative expression level of each *AcGA2ox* gene was calculated based on the comparison threshold period (2^−ΔΔCt^) method [47]. Pineapple actin-101 gene was used as the internal reference gene. Relative quantitative analysis was performed on the data obtained, and statistical analysis was performed using EXCEL 2021 and Graph Pad Prism 8.0 software.

### 4.8. Subcellular Localization of GA2ox Gene Family in Pineapple

Seven gene design primers were selected from the four clades of AcGA2ox, the cDNA of pineapple was cloned and the bands of correct size (TIANGENDP219-03) were recovered using a kit, and the primer sequences are shown in (Appendix A). The vector was enzymatized with SmaI infusionDNA, ligase was used to connect to the Ubi-GFP (Qin-Lab, Fuzhou, China) carrier and the obtained ligate was transformed into DH5α receptor cells. After PCR amplification and sequencing, positive clones were screened, and the plasmid was extracted to obtain the fusion expression vector Ubi:AcGA2ox:GFP and the target gene. The constructed vector plasmid was transferred to Agrobacterium GV3101 by electrotransformation method and cultured at 30 °C for 2d; Agrobacterium was suspended from solid to liquid medium of 10 mL LB, the suspensions were suspended and the activated bacterial solution OD600 ranged from 0.6–1.0. The tobacco infective liquid (MES 10 mM, MgCl_2_ 10 mM and AS 0.2 mM) was injected into the lower epidermis of the tobacco leaf, cultured in low light for 2d, prepared into the slide, observed and photographed by Leica-SP8 laser confocal microscope.

## 5. Conclusions

In this study, 42 GA2ox gibberellin oxidase genes were identified in pineapple, which were divided into 4 subfamilies. GA2ox is related to plant response to exogenous hormone expression and abiotic stress expression, and the expression level of GA2ox is different in different tissues. The main function of GA2ox is to inactivate the active bioactive gas GA. The results of this study provide a basis for further study on the evolution and function of the AcGA2ox gene family in pineapple plants.

## Figures and Tables

**Figure 1 plants-12-02673-f001:**
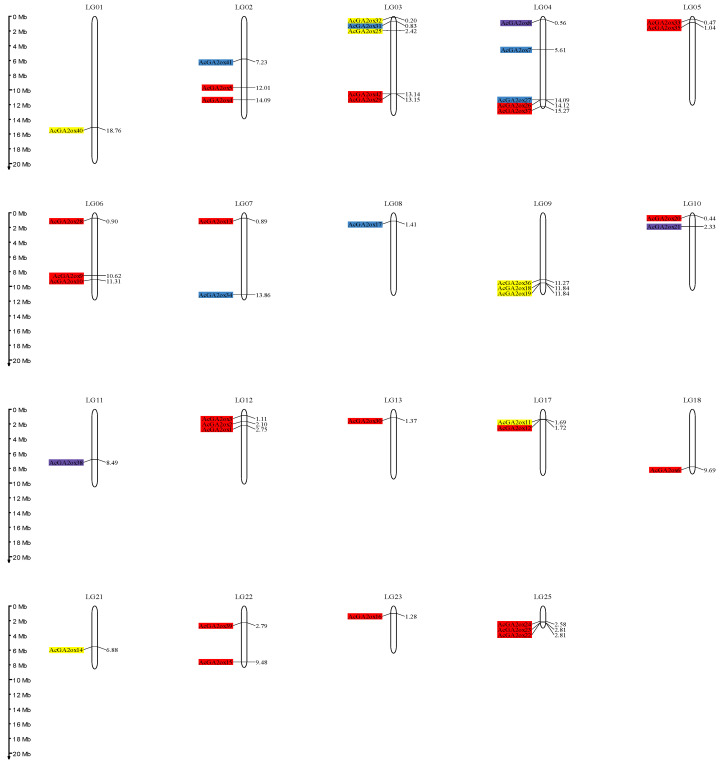
Genomic distribution of *AcGA2ox* genes on pineapple chromosomes. Note: Only the chromosomes bearing *AcGA2ox* genes are represented. Red: Clade I. Yellow: Clade II-A. Purple: Clade II-B. Blue: Clade II-C.

**Figure 2 plants-12-02673-f002:**
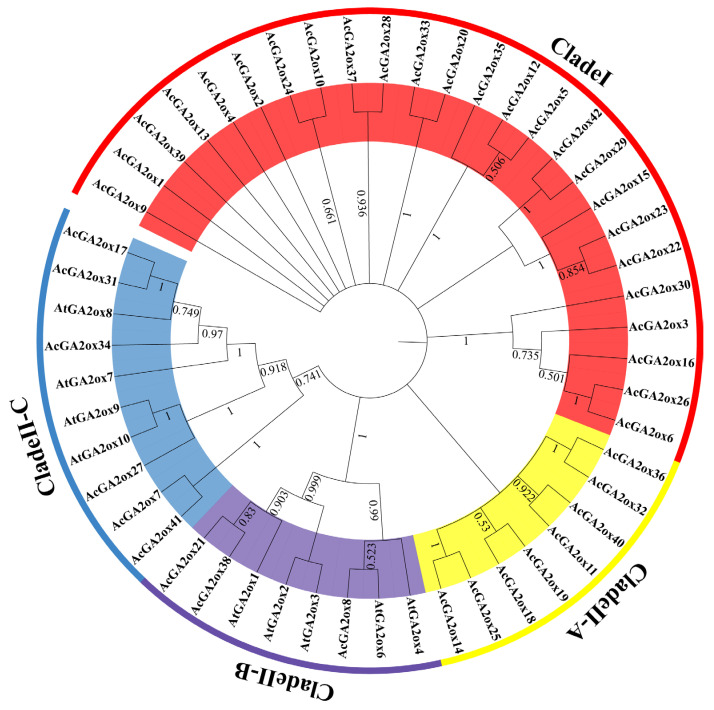
Phylogenetic analysis of GA2ox proteins from pineapple and *Arabidopsis*: The pineapple GA2ox proteins are classified into four groups: Clade I, Clade II-A, B and C.

**Figure 3 plants-12-02673-f003:**
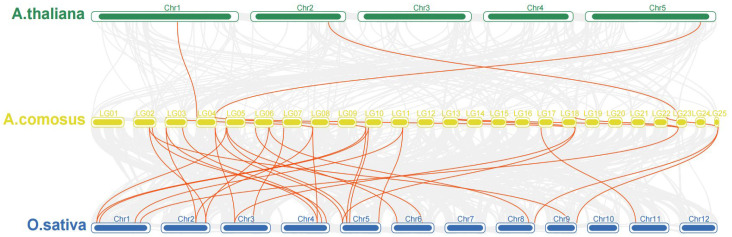
Collinearity analysis of Ga2ox family genes between *A. thaliana*, *A. comosus* and *O. Sativa*.

**Figure 4 plants-12-02673-f004:**
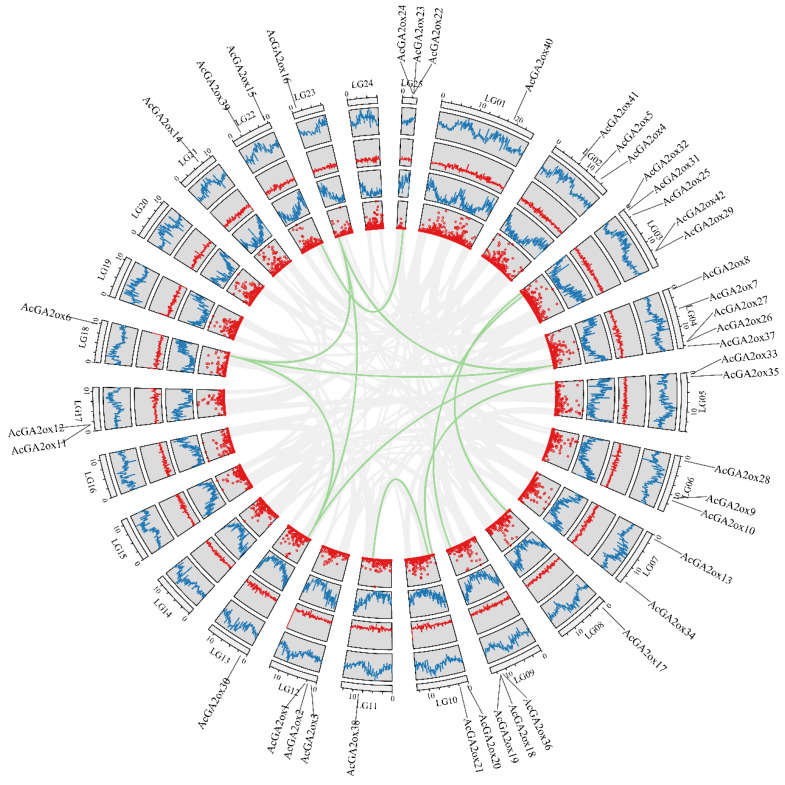
Collinearity analysis between *AcGA2ox* genes: The outer circle shows the distribution of the N ratio, the density of the GC skew gene and the change in the GC ratio of pineapple genome from inside to outside, respectively.

**Figure 5 plants-12-02673-f005:**
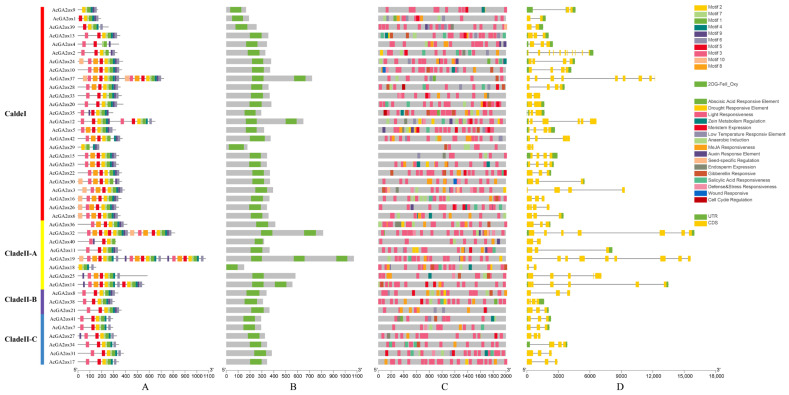
Gene structure and cis-acting element analysis of conserved motifs of the *AcGA2ox* gene family: (**A**) Motif composition of pineapple GA2ox proteins pattern of numbers 1–10 is displayed in boxes of different colors. Sequence information for each motif is provided in attached Appendix A. The length of protein can be estimated by following the scale. (**B**) The 2OG-Fell_Oxy domain is highlighted with a green box. (**C**) The promoter sequences of 42 *AcGA2ox* genes were analyzed with TBTools software (−2000 bp). According to the scale, the upstream length to the translation starting point can be inferred. (**D**) Exon–intron structure of the *AcGA2ox* gene green box indicates exon; black line indicates intron; and yellow box indicates the upstream/downstream region of *AcGA2ox* gene.

**Figure 6 plants-12-02673-f006:**
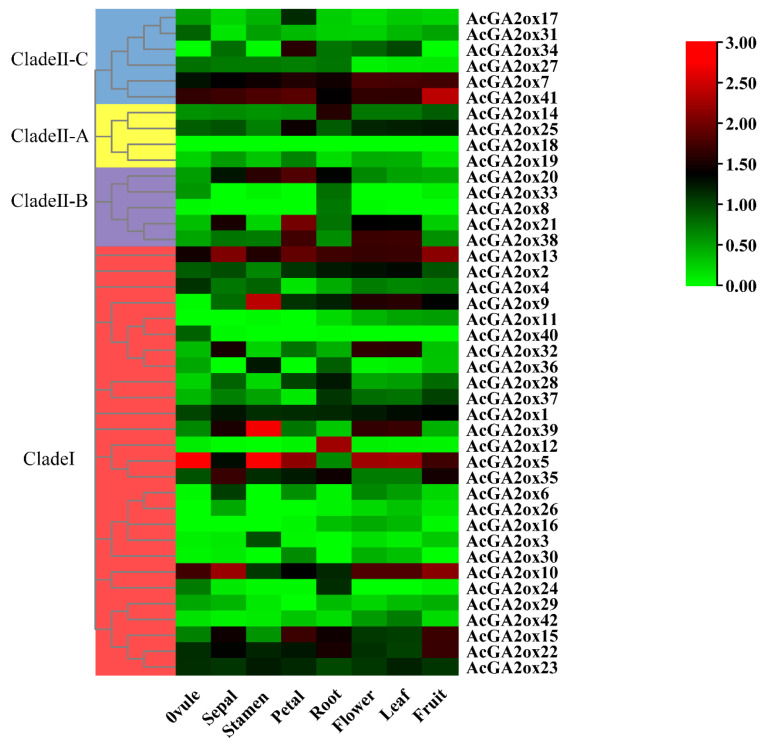
Expression profile of *GA2ox* genes in eight pineapple tissues (ovule, sepal, stamen, petal, root, flower, leaf and fruit).

**Figure 7 plants-12-02673-f007:**
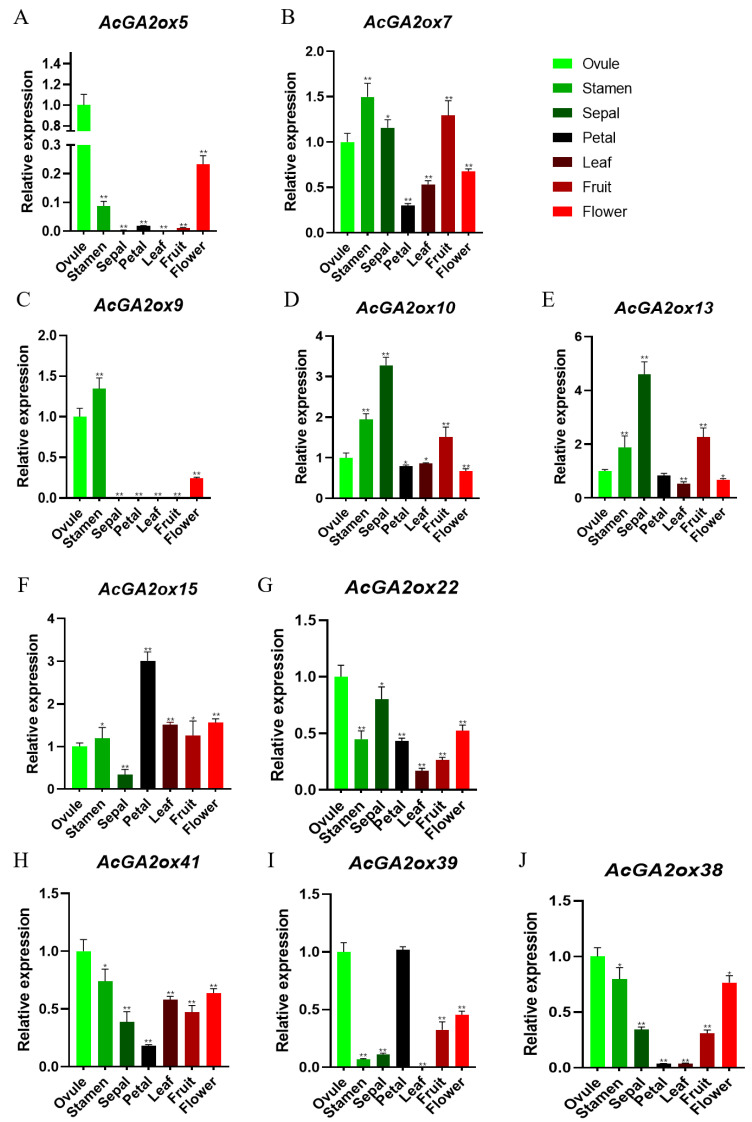
The expression of 10 *AcGA2ox* genes in 7 tissues was analyzed by qRT-PCR: Vertical bar indicates standard deviation. The asterisk indicates that the corresponding genes are significantly upregulated or downregulated compared with the untreated control group (* *p* < 0.05, ** *p* < 0.01 and Student’s *t*-test).

**Figure 8 plants-12-02673-f008:**
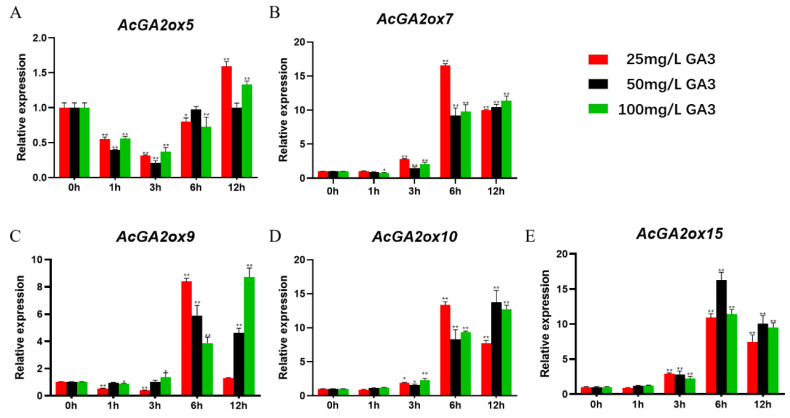
Expression levels of *AcGA2ox* in grape leaves after 12 h under different treatments with CK (contrast check): The expression level of the 10 *AcGA2ox* were genes acquired by qRT-PCR. Error bars indicate the standard deviation. Asterisks on top of the bars indicate statistically significant differences between the stress and counterpart controls (* *p* < 0.05, ** *p* < 0.01 and Student’s *t*-test).

**Figure 9 plants-12-02673-f009:**
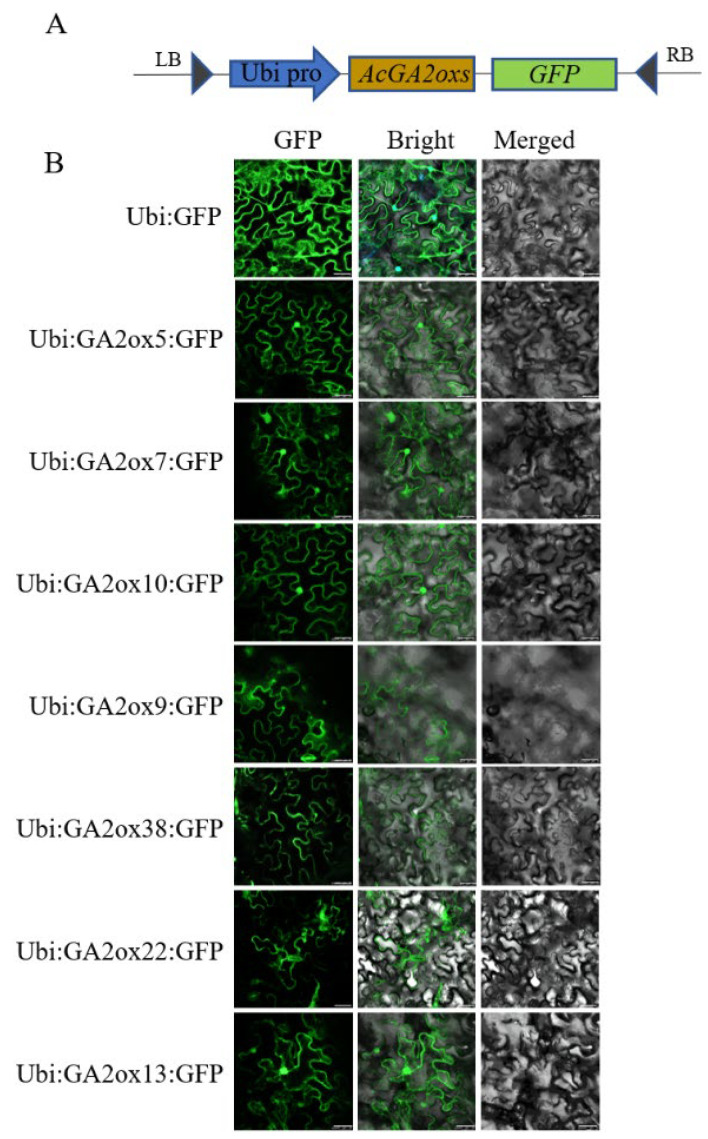
Subcellular localization analysis: (**A**) Construction pattern diagram of expression vector. (**B**) The fusion proteins Ubi:GFP and Ubi:AcGA2ox:GFP were transiently expressed in Nicotiana benthamiana leaf cells and observed with a laser scanning confocal microscope. Scale bar = 25 µm.

**Figure 10 plants-12-02673-f010:**
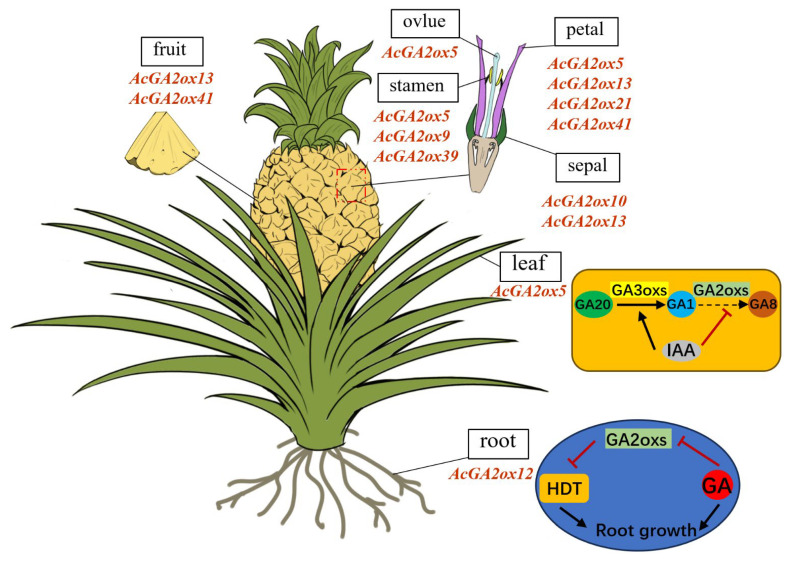
The potential function of *GA2ox* genes in pineapple: Schematic model of *AcGA2ox* gene expression patterns during leaf and root development. The *AcGA2ox* genes listed were highly expressed genes in corresponding tissues.

**Table 1 plants-12-02673-t001:** Distribution of *AcGA2ox* genes on chromosomes.

Chromosome	Number	Name	Chromosome	Number	Name
LG01	1	*AcGA2ox40*	LG14	0	
LG02	3	*AcGA2ox4*, *AcGA2ox5*, *AcGA2ox41*	LG15	0	
LG03	5	*AcGA2ox25*, *AcGA2ox29*, *AcGA2ox31*, *AcGA2ox32*, *AcGA2ox42*	LG16	0	
LG04	5	*AcGA2ox7*, *AcGA2ox8*, *AcGA2ox26*, *AcGA2ox27*, *AcGA2ox37*	LG17	2	*AcGA2ox11*, *AcGA2ox12*
LG05	2	*AcGA2ox33*, *AcGA2ox35*	LG18	1	*AcGA2ox6*
LG06	3	*AcGA2ox9*, *AcGA2ox10*, *AcGA2ox28*	LG19	0	
LG07	2	*AcGA2ox13*, *AcGA2ox34*	LG20	0	
LG08	1	*AcGA2ox17*	LG21	1	*AcGA2ox14*
LG09	3	*AcGA2ox18*, *AcGA2ox19*, *AcGA2ox36*	LG22	2	*AcGA2ox15*, *AcGA2ox39*
LG10	2	*AcGA2ox20*, *AcGA2ox21*	LG23	1	*AcGA2ox16*
LG11	1	*AcGA2ox38*	LG24	0	
LG12	3	*AcGA2ox1*, *AcGA2ox2*, *AcGA2ox3*	LG25	3	*AcGA2ox22*, *AcGA2ox23*, *AcGA2ox24*
LG13	1	*AcGA2ox30*			

**Table 2 plants-12-02673-t002:** Number distribution of GA2ox proteins.

Species	Clade I	Clade II-A	Clade II-B	Clade II-C
*Arabidopsis thaliana*	0	0	5	4
*Ananas comosus*	25	8	3	6

## Data Availability

All data analyzed during this study are included in this article and its Appendix A.

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
