# Peer review of "Identification of GA2ox Family Genes and Expression Analysis under Gibberellin Treatment in Pineapple (Ananas comosus (L.) Merr.)"

_plants, 2023, doi:10.3390/plants12142673_

Round 1

Reviewer 1 Report

Review: This paper presents a comprehensive analysis of the GA2ox gene family in pineapple, an important regulator of plant growth and development. The authors employed bioinformatics tools to identify and classify 42 AcGA2ox genes in the pineapple genome, analyzing their gene and protein structure, protein motifs, chromosome distributions, and expression profiles. The results of this study provide a valuable resource for further studies on the evolution and function of the AcGA2ox gene family in pineapple plants. 

The contributions of this paper include:

- Identification and characterization of 42 GA2ox genes in the pineapple genome, laying the foundation for further studies on the role of these genes in pineapple growth and development.

- Classification of AcGA2ox genes into four subgroups based on phylogenetic analysis, and analysis of their gene and protein structure, protein motifs, chromosome distributions, and expression profiles.

- Provision of useful information for further functional analysis of each member of the AcGA2ox gene family, potentially promoting the improvement of pineapple varieties through gene transfer technology.

Limitations of this study are:

- The study focuses solely on the identification and characterization of the GA2ox gene family in pineapple without investigating the functional roles of these genes in detail.

- The identified 42 AcGA2ox genes in the pineapple genome is relatively high compared to the nine AtGA2ox genes in Arabidopsis. The authors must explore why the pineapple has expanded this gene family. Moreover, when discussing the possibility that these newly identified pineapple-specific genes might have evolved new biological functions, the authors need to provide more substantial evidence to support this hypothesis.

- The authors have classified pineapple GA2ox genes into four distinct groups according to the phylogenetic analysis. However, the basis for this classification should be clearer, given the variety of potential classification methods. The method used for phylogenetic analysis is Neighbor-Joining (NJ), a widely used approach for constructing phylogenetic trees. Despite this, the NJ method has specific limitations that should be acknowledged. It can be sensitive to the choice of evolutionary distance metric, and the accuracy of the constructed tree may be compromised by mutations that are not proportional to evolutionary time, such as long branch attraction artifacts. Additionally, NJ does not provide explicit statistical measures or confidence estimates, such as bootstrap values, which are essential for evaluating the robustness of the inferred tree topology. Given these limitations, employing additional methods to construct phylogenetic trees for comparison, such as maximum likelihood (ML), would strengthen the reliability of their findings. 

- This study presents data on the expression of several GA2ox genes in different tissues of pineapple, but the methods used for expression analysis are poorly described. As detailed in the methodological section, it appears that only technical replicates were performed in the qPCR experiments, rendering the conclusions questionable. The study claims to have used qRT-PCR to study the relationship between GA2ox and GA in pineapple treated with different concentrations of gibberellin, but the sample preparation and expression data normalization method remain unclear. Furthermore, the study claims that the expression of some GA2ox genes was upregulated following GA3 treatment, but fails to present any time-course data to substantiate this claim. 

- Many figure captions lack detailed descriptions. Hence, more comprehensive captions are necessary.

- The tissue-specific expression patterns discussed in section 2.5 are valuable, as they provide insights into the potential biological roles of different AcGA2ox genes in pineapple growth and development. However, a more detailed explanation of the context of the observed tissue-specific expression patterns and the biological significance of these patterns for pineapple development or adaptation to different environments would enhance the value of these results for the scientific community.

- The experiment involving exogenous GA3 treatment and qRT-PCR analysis in section 2.6 is engaging. However, the authors should conduct controls and replicates to address any potential effects of exogenous treatments on gene expression patterns.

In conclusion, while this study presents some intriguing findings on the GA2ox gene family in pineapple, several issues need to be addressed regarding the methods and results. The methods employed for gene identification, expression analysis, and subcellular localization should be better described, and the study needs to present more data to support its claims.

Author Response

Point 1: - The identified 42 AcGA2ox genes in the pineapple genome is relatively high compared to the nine AtGA2ox genes in Arabidopsis. The authors must explore why the pineapple has expanded this gene family. Moreover, when discussing the possibility that these newly identified pineapple-specific genes might have evolved new biological functions, the authors need to provide more substantial evidence to support this hypothesis.

Response 1:The genome of Arabidopsis is small, and pineapple has a low retention rate during species evolution. We are using bioinformatics methods to explore pineapple GA2ox Gene family in order to find more potential similar domain Gene members of 2OG Well_ Oxy, some of which may be Pseudogene or repetitive events.

Point 2: - The authors have classified pineapple GA2ox genes into four distinct groups according to the phylogenetic analysis. However, the basis for this classification should be clearer, given the variety of potential classification methods. The method used for phylogenetic analysis is Neighbor-Joining (NJ), a widely used approach for constructing phylogenetic trees. Despite this, the NJ method has specific limitations that should be acknowledged. It can be sensitive to the choice of evolutionary distance metric, and the accuracy of the constructed tree may be compromised by mutations that are not proportional to evolutionary time, such as long branch attraction artifacts. Additionally, NJ does not provide explicit statistical measures or confidence estimates, such as bootstrap values, which are essential for evaluating the robustness of the inferred tree topology. Given these limitations, employing additional methods to construct phylogenetic trees for comparison, such as maximum likelihood (ML), would strengthen the reliability of their findings. 

Response 2: Thank you for your suggestion. We have found that the phylogenetic analysis method used in many reports is the NJ method. If the algorithm needs to be changed together with subsequent analysis, I have added a bootstrap value to the evolutionary tree.

Point 3: - This study presents data on the expression of several GA2ox genes in different tissues of pineapple, but the methods used for expression analysis are poorly described. As detailed in the methodological section, it appears that only technical replicates were performed in the qPCR experiments, rendering the conclusions questionable. The study claims to have used qRT-PCR to study the relationship between GA2ox and GA in pineapple treated with different concentrations of gibberellin, but the sample preparation and expression data normalization method remain unclear. Furthermore, the study claims that the expression of some GA2ox genes was upregulated following GA3 treatment, but fails to present any time-course data to substantiate this claim. 

Response 3: We have described the sample preparation and processing more accurately in the latest manuscript. If necessary, we can provide the original data of qPCR results.

Point 4: - Many figure captions lack detailed descriptions. Hence, more comprehensive captions are necessary.

Response 4: Thank you for your suggestion. We have made revisions to the title of the manuscript to make it more suitable for our content.

Point 5: - The tissue-specific expression patterns discussed in section 2.5 are valuable, as they provide insights into the potential biological roles of different AcGA2ox genes in pineapple growth and development. However, a more detailed explanation of the context of the observed tissue-specific expression patterns and the biological significance of these patterns for pineapple development or adaptation to different environments would enhance the value of these results for the scientific community.

Response 5: In our latest manuscript, we speculate on the potential function of the tissue-specific expression of parts of pineapple and present it in the model diagram.

Point 6: - The experiment involving exogenous GA3 treatment and qRT-PCR analysis in section 2.6 is engaging. However, the authors should conduct controls and replicates to address any potential effects of exogenous treatments on gene expression patterns.

Response 6: We performed three biological replicates in the treatment experiment and three technical replicates for each biological replicate, and their average values were calculated.

Reviewer 2 Report

I read the manuscript titled “Identification and expression analysis of GA2ox gene family in 2 Pineapple (Ananas comosus (L.) Merr.)” and found interesting However the manuscript needs major revision before acceptance for publication. In this study the authors performed in silico analysis to identify 42 GA2ox 20 genes (AcGA2oxs) in the pineapple genome and then functional insights have been derived through bioinformatic ways, however no wet data has been provided for functional validation. Further, There are a lot of English language mistakes and many sentences are difficult to understand. Some of the mistakes are mentioned below:

Abstract, please rewrite the following sentence  "In addition, we also analyzed the developmental expression profile of AcGA2oxs in different tissues, and 41 of the 42 AcGA2oxs genes were detected in the selected tissues, among which AcGA2ox18 may not have been expressed in the selected tissues or may have been a pseudogene”.

Line 258 “were transiently expressed in in Nicotiana benthamiana” should be written as “were transiently expressed in Nicotiana benthamiana”.

Ine 265, space is missing in sentence “various plants[29-31].

Line 280 to 282, please insert full stop among different sentences like “AcGA2ox contains some cis-acting elements related to hormone meristem stress and cell cycle regulation These genes are also affected by GA regulation Interestingly, a considerable number of gene promoters do not contain gibberellin-related elements.”.

Line 308 to 311, the same proble of full stop again “The pineapple variety was "MD2"and was provided by Qin Lab (Genome and Bio-308 technology Center of Fujian Agriculture and Forestry University, China) It was grown in a greenhouse at 25 ℃ with 16 hours of illumination and 8 hours of dark treatment at a relative humidity of 70% for one month.

Line 373, what is “The 2-Δ Δ Ct” ?

Please revise the title “Identification and expression analysis of GA2ox gene family in Pineapple (Ananas comosus (L.) Merr.)”. what expression? Gene, protein etc and wet or in silico? There is no need to use two names (english and scientific name in title Pineapple (Ananas comosus (L.) Merr.). please remove one or revise.

Author Response

I read the manuscript titled “Identification and expression analysis of GA2ox gene family in 2 Pineapple (Ananas comosus (L.) Merr.)” and found interesting However the manuscript needs major revision before acceptance for publication. In this study the authors performed in silico analysis to identify 42 GA2ox 20 genes (AcGA2oxs) in the pineapple genome and then functional insights have been derived through bioinformatic ways, however no wet data has been provided for functional validation. 

Response 1: Pineapple is asexual and has a long growth cycle, making it difficult to study its functions. We used exogenous hormones to treat pineapple and explore the transient changes in its gene expression. If further exploration of its function is needed, a longer period of exogenous treatment is required to further analyze its phenotype.

Further, There are a lot of English language mistakes and many sentences are difficult to understand. Some of the mistakes are mentioned below:

Abstract, please rewrite the following sentence  "In addition, we also analyzed the developmental expression profile of AcGA2oxs in different tissues, and 41 of the 42 AcGA2oxs genes were detected in the selected tissues, among which AcGA2ox18 may not have been expressed in the selected tissues or may have been a pseudogene”.

Response 2: We are very sorry for these errors in the manuscript. In the revised version, we carefully revised the relevant sentences and invited professional English editors to review the article.

Line 258 “were transiently expressed in in Nicotiana benthamiana” should be written as “were transiently expressed in Nicotiana benthamiana”.

Response 3: We are very sorry for the error in the manuscript, and we have already revised this sentence in the revised version.

Ine 265, space is missing in sentence “various plants[29-31].

Line 280 to 282, please insert full stop among different sentences like “AcGA2ox contains some cis-acting elements related to hormone meristem stress and cell cycle regulation These genes are also affected by GA regulation Interestingly, a considerable number of gene promoters do not contain gibberellin-related elements.”.

Response 4: We are very sorry for the error in the manuscript, and we have already revised this sentence in the revised version.

Line 308 to 311, the same proble of full stop again “The pineapple variety was "MD2"and was provided by Qin Lab (Genome and Bio-308 technology Center of Fujian Agriculture and Forestry University, China) It was grown in a greenhouse at 25 â„ƒ with 16 hours of illumination and 8 hours of dark treatment at a relative humidity of 70% for one month.

Response 5: We have rewritten this paragraph in the manuscript and provided a more detailed description of the material handling.

Line 373, what is “The 2-Δ Δ Ct” ?

Response 6: 2- ΔΔ Ct is the q-rtPCR calculation formula, and we have standardized the description method for this section in subsequent revisions to make it easier for readers to understand.

Please revise the title “Identification and expression analysis of GA2ox gene family in Pineapple (Ananas comosus (L.) Merr.)”. what expression? Gene, protein etc and wet or in silico? There is no need to use two names (english and scientific name in title Pineapple (Ananas comosus (L.) Merr.). please remove one or revise.

Response 7: Thank you for your suggestion. We have made revisions to the title of the manuscript to make it more suitable for our content.

Reviewer 3 Report

Manuscript ID: plants-2305656

In this manuscript, the authors identified 42 potential GA2ox genes in the pineapple genome based on amino acid sequence similarities to Arabidopsis GA2ox genes. While the authors have undoubtedly put in considerable effort, the presentation of the research and the logical flow of the study make it challenging to comprehend the rationale behind the experiments and the validity of the claims.

The study hinges on the authors' conclusion that the 42 potential pineapple genes are GA2ox genes, as they share sequence similarities with Arabidopsis GA2ox genes. However, considering the phylogenetic and collinearity analyses performed, the pineapple candidate GA2ox genes differ significantly from those in Arabidopsis. It is unclear why the authors chose to compare the pineapple genes to the Arabidopsis genome initially.

Assuming these 42 sequences are indeed potential GA2ox genes, there is no certainty that all of them are functionally active GA2ox genes. The authors' gene expression data suggest that most of the 42 genes are not even expressed.

The subcellular localization experiment utilizing a constitutively expressed promoter is insufficient for studying gene function. Ideally, the native promoter of each gene should be fused with GFP and expressed alongside the respective genome sequence. Furthermore, even if conducted appropriately, subcellular localization alone does not provide substantial evidence for the claim that “GA2ox functions in both the nucleus and the cell membrane and that its encoded products may be involved in the stabilization of the plasma membrane or the control of protein transport” (Line 254-256).

The manuscript's writing presents several issues. Firstly, the experimental methods are inadequately described. For instance, the RNA extraction process is unclear, and the referenced previous paper (reference 43) uses different tissues and stages than those reported in this manuscript. Secondly, the introduction is overly general and lacks the necessary information to prepare readers for the article, while the discussion is superficial.

The authors have been careless in manuscript preparation, resulting in numerous grammar mistakes, incorrect symbol usage, and unclear labeling of acronyms. 

Author Response

In this manuscript, the authors identified 42 potential GA2ox genes in the pineapple genome based on amino acid sequence similarities to Arabidopsis GA2ox genes. While the authors have undoubtedly put in considerable effort, the presentation of the research and the logical flow of the study make it challenging to comprehend the rationale behind the experiments and the validity of the claims.

Response 1: At present, there is no publicly available pineapple database. Arabidopsis and rice are model crops, and the entire omics sequence has been determined, providing reference for analyzing other species.

The study hinges on the authors' conclusion that the 42 potential pineapple genes are GA2ox genes, as they share sequence similarities with Arabidopsis GA2ox genes. However, considering the phylogenetic and collinearity analyses performed, the pineapple candidate GA2ox genes differ significantly from those in Arabidopsis. It is unclear why the authors chose to compare the pineapple genes to the Arabidopsis genome initially.

Response 2: We are using bioinformatics methods to explore pineapple GA2ox Gene family in order to find more potential similar domain Gene members of 2OG Well_ Oxy, some of which may be Pseudogene or repetitive events.

Assuming these 42 sequences are indeed potential GA2ox genes, there is no certainty that all of them are functionally active GA2ox genes. The authors' gene expression data suggest that most of the 42 genes are not even expressed.

Response 3: We have made modifications to this conclusion and explained its potential functions in the discussion.

The subcellular localization experiment utilizing a constitutively expressed promoter is insufficient for studying gene function. Ideally, the native promoter of each gene should be fused with GFP and expressed alongside the respective genome sequence. Furthermore, even if conducted appropriately, subcellular localization alone does not provide substantial evidence for the claim that “GA2ox functions in both the nucleus and the cell membrane and that its encoded products may be involved in the stabilization of the plasma membrane or the control of protein transport” (Line 254-256).

Response 4: We have made modifications to this conclusion and explained its potential functions in the discussion.

The manuscript's writing presents several issues. Firstly, the experimental methods are inadequately described. For instance, the RNA extraction process is unclear, and the referenced previous paper (reference 43) uses different tissues and stages than those reported in this manuscript. Secondly, the introduction is overly general and lacks the necessary information to prepare readers for the article, while the discussion is superficial.

Response 5: We have referred to the processing methods in more reports and made modifications in the text.

The authors have been careless in manuscript preparation, resulting in numerous grammar mistakes, incorrect symbol usage, and unclear labeling of acronyms. 

Response 6: We are very sorry for these errors in the manuscript. In the revised version, we carefully revised the relevant sentences and invited professional English editors to review the article.

Round 2

Reviewer 1 Report

I have carefully reviewed the revised manuscript titled "Identification and expression analysis of GA2ox gene family in Pineapple" by Zhu and colleagues. I am pleased to report that the authors have meticulously addressed all the issues raised in the first round of reviews.

The improvements made to the manuscript have greatly enhanced its clarity, precision, and overall quality. The authors have revised the analysis in response to the comments made during the initial review, and they have done an excellent job integrating these new insights into the work. The manuscript now provides a comprehensive and coherent examination of the GA2ox gene family in Pineapple, offering valuable contributions to the field.

In conclusion, I believe the manuscript has been significantly improved and is now worthy of publication in Plants. I commend the authors for their diligent work on these revisions.

Reviewer 2 Report

The manuscript has been improved enough and I would like to recommend its publication in its current form. 

Reviewer 3 Report

I'm impressed with the revised version, and the authors made some changes that addressed my questions and/or concerns. I understand that researches in  non-model species are difficult. I think this version of manuscript contains correct information with proper descriptions.